# Oral Immunotherapy for Children with Cow’s Milk Allergy: A Practical Approach

**DOI:** 10.3390/children9121872

**Published:** 2022-11-30

**Authors:** Maria Angela Tosca, Roberta Olcese, Guido Marinelli, Irene Schiavetti, Giorgio Ciprandi

**Affiliations:** 1Allergy Center, IRCCS Giannina Gaslini, 16100 Genoa, Italy; 2Health Science Department, University of Genoa, 16100 Genoa, Italy; 3Allergy Clinic, Casa di Cura Villa Montallegro, 16100 Genoa, Italy

**Keywords:** milk allergy, oral immunotherapy, desensitization, tolerance, adverse reaction, children

## Abstract

Cow milk allergy (CMA) is a prevalent disease in childhood. Natural history is usually favorable as CMA can disappear by school age in many subjects. Diagnosis corresponds to treatment, as an elimination diet is a solution. However, cow’s milk (CM) is real food, hardly replaceable. Thus, CM reintroduction represents a demanding challenge in clinical practice. The induction of CM tolerance could be achievable using oral immunotherapy (OIT), such as the administration of increasing milk quantities until reaching tolerance. However, the OIT schedule and procedure need to be better standardized, and performance may vary widely. Therefore, the present study reports the practical experience of a third-level pediatric allergy center in managing children with CMA and submitting them to OIT. OFC and OIT are relatively safe procedures as the reaction rate is low. Almost two-thirds of the OIT subjects tolerated CM. Reactions were associated with high IgE levels. Therefore, the present experience, developed by a qualified center, may suggest and propose a practical approach for managing children with CMA. After the initial workup, including a thorough history, physical examination, and laboratory tests, OFC and, when indicated, OIT could be performed in most children with CMA.

## 1. Introduction

Cow milk allergy (CMA) is a prevalent problem in childhood. However, the worldwide CMA prevalence may significantly vary, mainly concerning the criteria used to define it. Two main measures are used in epidemiological surveys: true CMA, such as with a validated diagnosis, or self-reported CMA, obviously approximate.

The prevalence of self-reported CMA ranges from 1 to 17.5% in preschoolers, 1 to 13.5% in schoolers, and 1 to 4% in adults [1]. However, the prevalence of true CMA, when provided by well-documented studies, ranges from 0.21 to 4.9% in childhood [2,3,4,5,6]. In addition, a UK study monitored 13 million children and reported 23 cases of anaphylaxis due to CMA [7]. Thus, CMA is a common disease that may also be highly dangerous.

The main cow’s milk allergenic molecules include α-lactalbumin (Bos d 4), β-lactoglobulin (Bos d 5), and casein (Bos d 8); the sensitization rate widely varies from 0 to 88%, while allergy occurs in about 20% of the sensitized subjects [8].

Cross-reactions among milk allergens from different mammalian species are of clinical relevance. The most significant homology occurs among the milk proteins of cows, sheep, and goats such as *Bos* (cattle), *Ovis* (sheep), and *Capra* (goats), which belong to the ruminant bovine family. The proteins in their milk possess little structural similarities with those of the *Suis* (pig), *Equinae* (horse and donkey), and *Camelidae* (camel and dromedary), and also with human milk. 

Notably, milk allergens preserve their biological activity even after boiling, pasteurization, ultra-high temperature treatment, evaporation, or milk powder production for infants. Therefore, to obtain hypoallergenic formulas, extensive hydrolysis is required in addition to further processing, such as heat treatment, ultrafiltration, and pressurization application. Attempts have been made to classify the formulas into partially and extensively hydrolyzed products according to their degree of protein fragmentation. Still, there is yet to be an agreement on the criteria for this classification. However, hydrolyzed formulations, so far, are a valuable source of widely used proteins for infants with CMA, even if they still have poor palatability.

The CMA natural history is characteristic, as it tends to onset before 12 months of age concurrently with the CM introduction. However, the prognosis is usually good as most children acquire tolerance to milk over time [2]. Namely, symptoms typically develop early after the first CM ingestions. Even if it may improve in most subjects by school age, some may have persistent CMA until adulthood [9,10]. 

Tolerance to CM, evaluated with an oral food challenge, is the primary parameter to evaluate allergy resolution or persistence. Consistently, cow’s milk tolerance is associated with lower allergen-specific IgE in comparison with higher levels in children with a positive challenge [11,12].

Risk predictors for persistent CMA are multiple allergies (particularly egg allergies), high IgE levels, asthma, allergic rhinitis, immediate symptoms upon milk intake, and reactions to milk in baked foods [13].

Several studies have shown that the IgE levels when measured by a skin test and/or serum assay, can discriminate between allergy persistence and future tolerance [9,10,14]. For example, a wheal diameter < 5 mm may predict tolerance in 83% by the age of four years. Consistently, high serum IgE levels predict CMA severity and persistence over time, while a downtrend may mean a favorable prognosis [2,5]. In addition, increased levels of allergen-specific IgE predict the risk of anaphylaxis during the oral milk challenge [15]. 

Immediate symptoms after CM ingestion involve mainly the skin, gastrointestinal, and respiratory tract. Cardiovascular symptoms are rare. Symptoms may range from mild to severe to anaphylactic shock, which can also be fatal [16,17].

Food protein-induced enterocolitis syndrome (FPIES) is a severe manifestation; symptoms are acute and often severe, including profuse vomiting within 1–4 h of milk intake, or diarrhea within 4–10 h, lethargy (65–100%), pallor (30–90%), and hypothermia [18]. In addition, dysphagia may suggest eosinophilic esophagitis, but the diagnosis requires a biopsy [19]. Initially, FPIES was classified within the CMA phenotypes, but presently, the pathogenetic mechanisms recognize non-IgE-mediated pathways.

Considering the potential severity of CMA and dietetic implications, an early diagnosis is important to initiate an appropriate diet and avoid unnecessary dietary restrictions. In addition, the diagnosis of CMA requires consistency between history and sensitization to CM allergen, documented by a skin test and/or serum assay. In other words, symptoms should occur after ingestion of the sensitizing allergen.

International guidelines provide cut-offs for the skin test and serum assay that are helpful for discriminating between sensitization and allergy in clinical practice [20,21,22,23,24,25,26]. However, IgE tests may not be complete diagnostic criteria, and a placebo-controlled oral food challenge (PC-OFC) remains the gold standard for diagnostic confirmation [20,21,22,23,24,25,26]. Still, it is laborious and time-consuming, so OFC with CM and without a placebo is the standard in clinical practice. The oral challenge can be performed with products containing milk, dairy products, or fresh milk. Bakery products with milk as an ingredient are usually less allergenic, even though it is not definitely proven for casein [26]. So, they are commonly used in cases of severe milk reactions. 

Once a diagnosis has been made, treatment must be considered. Treatment for CMA would be straightforward as it consists of avoiding cow’s milk and products containing it. Moreover, the family and caregivers of the child (including school personnel) should be informed about the presence of milk as an ingredient or traces (in severe cases) in the diet. Allergy follow-up is recommended to evaluate the reintroduction of milk and dietary follow-up to assess the patient’s nutritional status, as nutritional deficiency may occur. However, an elimination diet is tough as CM is an ingredient in many industrial preparations. Under two years of age, in the case of a non-breastfed child who is allergic to milk, it is mandatory to propose a milk substitute, mainly including adapted formula [20,21,22,23,24,25,26,27].

The CM reintroduction represents a compelling task and deserves attention and patience from doctors and parents (caregivers). The dilemma is when to reintroduce CM, as cow’s milk is an essential food in infancy and is necessary for adequate growth. As a result, CM reintroduction remains a crucial issue in clinical practice.

Oral immunotherapy, such as reintroducing the offending allergen, is an intensely investigated medical procedure. The OIT aim is to induce immunological and clinical tolerance to the causal food allergen, as in the immunotherapy for inhalant allergens [20,21,22,23,24,25,26,27]. Although the guidelines emphasize the relevance of standardization of OIT procedures, in clinical practice, the OIT protocols can be different, reflecting personal experience and local necessity. As a result, much effort still needs to be made to make OIT for CMA more accessible and widespread.

Based on this background, the present study aimed to report the experience of a third-level allergy center in managing children with CMA and undergoing OFC and OIT for CMA.

## 2. Materials and Methods

The present study retrospectively evaluated all children admitted to performing OFC and possibly OIT for CMA from 2012 to 2021. The data were extracted from an electronic platform containing demographics, clinical history, in vitro-laboratory, and in vivo tests; the data related to clinical parameters and the monitoring of the patients during oral food challenges and OIT procedures with a close follow-up. In particular, the dose of tolerated CM, the occurrence of adverse reactions, and the use of symptomatic medications were reported.

Laboratory data included peripheral eosinophils, total serum IgE, and allergen-specific IgE to casein, α-lactalbumin, β-lactoglobulin, and raw milk. The data were stored in an electronic database.

The measurement of IgE was performed using ELISA assays provided by Thermofisher (Milan, Italy). The total serum IgE normal level is <60 kU/mL for 1–5-year-old children, <90 kU/mL for 5–9-year-old children, and <100 kU/mL for older children. Sensitization is defined when the value of allergen-specific IgE is >0.35 kUA/mL.

The inclusion criteria were suspected CMA, suggested by symptom occurrence after CM ingestion or the previous detection of sensitization to CM, and pediatric age. The suspected allergic symptoms consequent to CM ingestion included urticaria, angioedema, oral allergic syndrome, gastrointestinal symptoms (gastralgia, nausea, vomiting, colic, diarrhea), symptoms of a systemic reaction (anaphylaxis), and the exacerbation of atopic dermatitis. Exclusion criteria were concomitant diseases (e.g., acute urticaria) and medications (e.g., concurrent use of systemic corticosteroids) that, in the doctor’s opinion, could interfere with interpreting results.

The oral food challenge was performed using three different schedules: A “low-dose and slow” schedule: patients started OFC by taking an initial dose of 0.05 mL (one drop) of CM; this dose contained 1.7 mg of CM protein. Alternatively, a quantity of biscuit (Plasmon biscuit) or cheese (Parmesan) containing the same amount of CM protein was used. The initial dose was increased by one drop at a time every 30 min if the previous dose was well tolerated. The procedure comprised a maximum of seven administrations per day. The challenge ended when a dose of 1 mL of CM was reached or if the subject presented with a clinical reaction. Usually, the challenge procedure lasts 2–3 days.A “rapid incremental” schedule: patients started from initial doses of 0.05 mL (one drop) of CM. The dose was doubled every 30 min if the last dose was well tolerated. The OFC end reached a cumulative dose of between 1 mL and 40 mL of CM. Usually, the challenge procedure lasts two days.An “ultra-rash” schedule: the starting dose was 1 mL of CM. The build-up phase was very quick as the dose was doubled every 30 min if the last dose was tolerated. For this schedule, only CM was used. The final dose was more than 40 mL of CM.

The criteria for choosing the schedule were based on the history of the severity of the symptoms following the intake of CM. The “low-dose and slow” schedule concerned children with severe CMA. The “rapid incremental” schedule concerned children with mild-moderate CMA. The “ultra-rash” schedule concerned only children with sensitization to CM.

The oral immunotherapy was performed using three different schedules:“Low-dose and slow build-up”: the patient performed slow and gradual increases, starting with an initial dose of CM (or an equivalent quantity of CM proteins contained in a cookie or cheese), which was less than 33% of that obtained at the challenge. The dose was taken daily. The increase, usually of 10%, occurred monthly.“Rapid incremental build-up”: the initial dose ranged between 33% and 66% of that obtained at the challenge. This dose lasted for 1–2 weeks. Successively, the tolerated dose increased by 20% each week.“Ultra-rash build-up”: the initial dose was >66% of that obtained at the challenge. This dose was maintained for a week. Successively, the dose increased by 50% every week.

The build-up period usually lasted from 3 to 9 months. Successively, the dose was maintained at home for months to years and usually was about 300–400 mg of CM proteins/day. An oral food challenge was repeated every 6–12 months to assay the tolerance.

The selection criteria for the OIT schedule were the same as those used for OFC.

The Ethics Committee of the IRCCS Istituto Giannina Gaslini approved the procedure (code number: 22253/2017). Parents signed informed consent that detailed all the information needed to understand what the procedure consists of, the risks and benefits, and the rules that had to be followed in the hospital environment and at home.

The data are expressed as numbers, percentages, and standard deviations. The statistical analysis was performed using the non-parametric Mann–Whitney U test. The significance level was set at 0.05 SPSS v.24 (SPS, Bologna, Italy) and was used for computation. 

## 3. Result

### 3.1. General Characteristics

Seventy-six children, 54 males (71.1%) and 22 females (28.9%), with a mean age of 45.3 months, performed OFC to CM. A sensitization to inhalant allergens was documented in 51 (67.1%) children, and sensitization to the food allergen was performed in 43 (56.6%) children. The most common comorbidity was asthma (39 patients, 51.3%), followed by atopic dermatitis (31 patients, 40.8%), allergic rhinitis (21 patients, 27.6%), urticaria (5 patients, 6.6%), and oral allergic syndrome (1 patient, 1.3%), as reported in Figure 1. 

### 3.2. Clinical Characteristics

Two groups of patients were identified: (i) 10 (13.2%) patients with a mere sensitization to CM proteins but without a clinical reaction after CM ingestion, and (ii) 66 (86.8%) patients with the clinical reaction after intake of raw milk, milk derivatives, or milk-containing foods. 

The first allergic reaction occurred under two years of age in 54 (81.8%) children, between 24 and 72 months of age in 9 (13.6%), and in school age (>6 years of age) in 3 (4.6%). As concerns clinical severity, 21 (31.8%) mild reactions, 21 (31.8%) moderate, and 24 (36.4%) severe were reported; 11 (16.7%) required adrenaline injection. The most common symptoms were: urticaria/rash (58 patients, 87.9%); abdominal pain, nausea, vomiting, and diarrhea (27, 40.9%), of which FPIES-like symptoms (1); nasal itching, nasal obstruction, rhinorrhea (20, 30.3%); cough, laryngeal stridor, wheezing (9, 13.6%); and weakness, hypotonia, hypotension, syncope (6, 9.1%). Forty (60.6%) patients had symptoms affecting two or more organs or systems (anaphylaxis).

Raw milk ingestion was the culprit in 36 patients (54.5%), milk derivatives/milk flours in 20 (30.3%), and milk contained as an ingredient in doughs in 10 (15.2 percent).

Regarding the clinical presentation, 51 patients (67.1%) presented a reaction since the first introduction of milk or milk derivatives; 15 (19.7%) developed a clinical response after an initial tolerance, and 10 (13.2%) tolerated milk and derivates in a little amount, without experiencing symptoms before the allergic reaction occurred for which they were evaluated.

### 3.3. Biological Test before OFC

Only three patients had eosinophilia, including >500 cells/μL, high total serum IgE levels, and increased allergen-specific IgE levels in all subjects (Table 1).

### 3.4. Characteristics of Patients Undergo to OFC 

The mean age was 69.5 (±45.3) months, ranging from 5.4 months to 10 years. OFC lasted 2.1 days (±0.99), ranging from 1 to 5 days.

Considering the three different procedures, eight patients (10.5%) performed the “low-dose and slow” schedule, fifty-one patients (67.1%) the “rapid incremental” schedule, and seventeen patients (22.4%) the “ultra-RASH” schedule.

The starting preparation was milk in 68 patients (89.5%), followed by a cookie (6, 7.9%) and cheese (2, 2.6%). The final preparation was milk in 70 (92.1%) patients, derivatives in 2 (2.6%), and foods containing milk (cookies) in 4 (5.3%). 

Immediately after OFC, 17 patients (22.4%) started CM OIT. The first dose corresponded to the total cumulative daily amount on the last OFC day. The mean starting dose was 25.6 (±37.5) mL, ranging from 0.1 to 150. OIT was conducted in the ward. Concerning the other patients, 13 patients (17.1%) were discharged on an unrestricted diet, 5 (6.6%) were discharged on a milk-free diet, and 2 (2.5%) had an FPIES diagnosis. On the whole, OIT was proposed to 57 patients; all parents agreed to submit their children to OIT.

### 3.5. Reactions during OFC

The OFC caused reactions in 31 patients (40.8%); 22 (71.5%) had mild reactions, 5 (16.1%) were moderate, and 4 (12.9%) were severe. Reactions to OFC involved the skin in 22 patients (71.0%), as reported in Figure 2, the gastrointestinal tract in 12 (38.7%), the upper respiratory tract in 7 (22.6%), the lower respiratory tract in 8 (25.8%), and the cardiovascular system in 1 (3.2%). In addition, OFC caused anaphylactic reactions in 11 (14.5%) patients, which were promptly resolved after the administration of intramuscular adrenaline. 

### 3.6. Characteristics of Patients Undergo to OIT 

Fifty-seven (75%) patients started OIT. The starting dose was less than 1 mL in 17 patients (32.7%); the remaining subjects started ingesting > 1 mL.

Fifty-three patients (93%) started taking milk, four (7%) milk derivatives, or foods containing milk (cookies/parmesan). The starting dose was usually maintained daily for a variable period of 1–4 weeks.

The OIT starting dose was calculated as the percentage of the cumulative dose administered on the last day of OFC. Consequently, 23 patients (40.4%) started with a dose < 33% of the last cumulative OFC dose; 20 (35.1%) started with a dose between 33 and 66%; 10 (17.5%) had a dose > 66%. Therefore, the mean value was 45.5%, corresponding to a median of 45.5 mL.

Regarding the OIT schedule, 14 patients (24.6%) performed the “low-dose and slow” schedule, 23 patients (40.4%) the "rapid incremental" schedule, and 20 patients (35.1%) the “ultra-RASH” schedule.

Seventeen patients (29.8%) temporarily interrupted OIT: 18 patients once, four twice, and one four times, but then they resumed the OIT procedure. However, six (10.5%) discontinued OIT because of severe symptoms after accidental milk intake.

Regarding safety, no patient in “Ultra-rash” OIT had moderate/severe reactions, and five had mild reactions. Only two patients (3.5%), following the “rapid-incremental” schedule, required adrenaline injections during the OIT starting phase, but anaphylactic symptoms were promptly resolved. 

Successively, during OIT, two patients (3.5%) had severe reactions after milk-contaminated food/accidental intake of milk, 34 (59.6%) had mild reactions during OIT, and six (21%) had mild reactions from accidental intake of milk. Accidental intake is defined as unintended and fortuitous. 

### 3.7. Last Follow-Up Visit of Patients Undergo to OIT

Seventeen patients (30%) were continuing OIT. Globally, 36 patients (63.2%) were taking milk regularly in the diet without presenting symptoms. However, nine (15.8%) preferred to avoid milk but were taking foods labeled ’may contain traces of milk" without problems, eight (14%) preferred to avoid milk and foods labeled ’may contain traces of milk", and four (7%) were lost to follow-up.

### 3.8. Biological Test at the Last Follow-Up

Eosinophilia and IgE levels did not change substantially after OIT (Table 2). 

Moreover, patients were stratified into two subgroups: patients with reactions during OIT and patients without reactions. Comparing serum IgE levels, IgE to casein, α-lactoalbumin, and raw milk were significantly higher in patients with reactions than in patients without reactions (*p* < 0.001, <0.01, and <0.001, respectively) as reported in Table 3.

## 4. Discussion

The management of children with CMA is still a demanding task in clinical practice. In addition, OIT procedures are not univocal, and fear of severe reactions significantly affects the actual availability of this causal treatment. Many protocols and formulas for OIT have been proposed, but there is no standard procedure that has been adopted everywhere at present. However, milk reintroduction is often necessary to guarantee adequate nutritional requirements and an acceptable quality of life.

The current study reported the practical approach to managing children with CMA and undergoing OFC and OIT. However, the results suggested that the CMA course is hugely variable and multifaceted. Consistently, OFC outcomes and OIT effects could vary among children. The findings reported that CMA was associated with a high frequency of skin reactions, followed by gastrointestinal and lower respiratory tract reactions. It should also be noted that 9.1% of patients developed cardiovascular symptoms. These results were consistent with a recent study conducted in Singapore showing the preeminence of cutaneous symptoms [28].

Moreover, the outcomes were encouraging as the prognostic perspectives were also favorable for patients with a history of severe reactions. Indeed, more than half of the subjects undergoing OIT could tolerate milk and, thus, follow an unrestricted diet. In addition, OIT was a substantially safe procedure, as only two anaphylactic reactions occurred during all introductions. However, these events underscore the need to perform the initial steps in a protected environment (in a hospital with trained staff) since two anaphylactic reactions occurred during OIT, and continuation at home could be performed in patients with less severe, non-anaphylactic reactions and with self-injected adrenaline anyway.

Compliance with the OIT procedure remained a critical issue as a fair number of children interrupted OIT, probably because the procedure was long and laborious. Notably, it has to be underscored that stopping the intake of foods already tolerated may be dangerous as tolerance can be lost, even quickly.

We also emphasize that allergy tests (the serum IgE assay for raw milk and molecules) should be monitored to understand when to perform OFC and OIT safely, as recently updated [29]. Consistently, OIT reactions were more frequent in children with higher levels. As a result, the OIT schedule should be slow and requires attention in the presence of high levels of specific IgE.

Many children tolerated an unrestricted diet after OIT. However, some parents were reluctant to give milk, but the children could still tolerate products containing traces. They would probably have tolerated baked milk products, as recently shown by the iAGE study [30].

Another intriguing result concerned the lack of the OIT effect on allergen-specific IgE in milk molecules. OIT probably acts on tolerogenic mechanisms, independent from a type 2 response, but involving Breg and Treg cells as actors of natural immunotolerance [31,32]. 

The present study had some limitations, including the retrospective design, the lack of quality of life assessment, and the relatively limited number of patients. However, the results reflect what happens in clinical practice, such as in real life. Managing children with CMA is a demanding task for pediatricians, taking into account the high prevalence of this medical condition. In this regard, a recent US study reported a prevalence of CMA of about 5% [33]. This high prevalence entails a relevant population-level burden with substantial healthcare costs, psychosocial relevance, and nutritional repercussions. Therefore, children with CMA deserve careful management. As a result, OFC and OIT constitute valuable opportunities that should be pursued. However, the initial management should be conducted in specialistic settings where the workup and therapeutic strategy can be adequately addressed. From a practical point of view, a demanding question concerns the OIT proposal in patients with a laboratory profile predictive of severe reactions. A clear-cut response does not exist. However, the risk can be severe if the subject is accidentally exposed to milk or milk derivatives, even sometimes in trace amounts, in a non-hospital setting, for example, in a school setting where they are otherwise unprotected, without any medical supervision or pharmacological safeguards.

Therefore, the results of the experience developed by a third-level allergy center could be summarized as follows: after the initial workup, including a thorough history, physical examination, and laboratory tests, OFC and, when indicated, OIT could be performed, as synthetically represented in Figure 3.

## 5. Conclusions 

The present study reported the experience of a third-level allergy center in managing CMA children. The obtained results suggest that OFC is crucial for the diagnosis of CMA but must be performed in a protected environment and by trained staff because anaphylactic reactions are not uncommon. However, prompt and adequate treatment enables their rapid resolution. 

Similarly, OIT should also be conducted in a protected setting, at least in the initial stages, particularly in children with a severe profile and during incremental challenges. Indeed, anaphylactic reactions can always occur and require prompt treatment.

The present experience also suggests that most children who performed OIT tolerated milk. However, adequate information regarding milk’s presence (even trace amounts) in food still needs to be emphasized. A fair percentage of children avoided milk intake anyway. In addition, appropriate education in the early recognition of signs of anaphylaxis and self-treatment with adrenaline is essential.

It also has to be underlined that a “low dose and slow” or “rapid incremental” were globally tolerated and could be preferred for most patients.

In conclusion, the present practical experience could positively contribute to supporting the practice of performing OIT whenever possible for children with a definite diagnosis of CMA to encourage a physiological intake of cow’s milk.

## Figures and Tables

**Figure 1 children-09-01872-f001:**
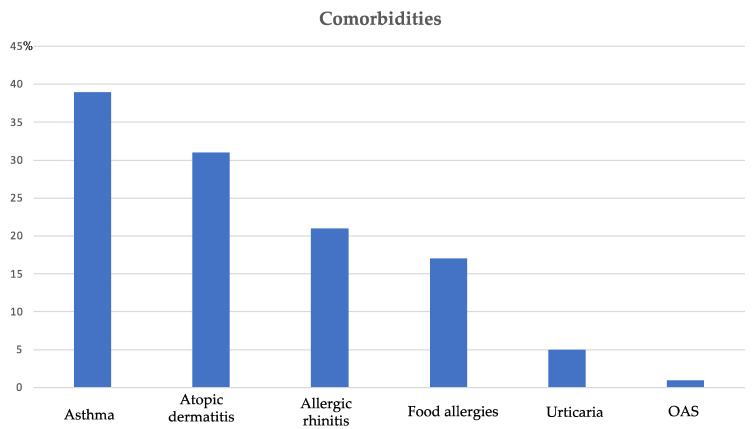
Comorbidities in patients performing OFC to cow’s milk.

**Figure 2 children-09-01872-f002:**
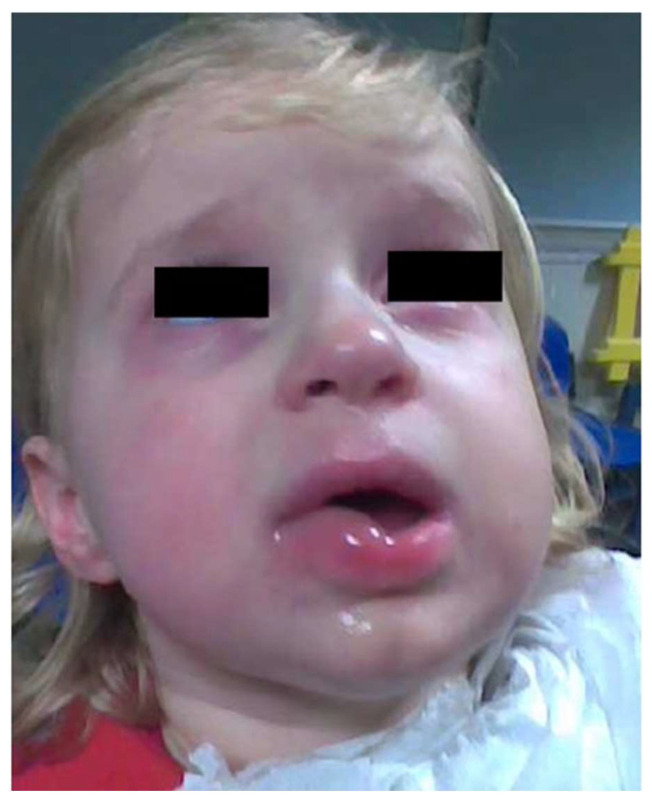
Mucocutaneous reaction during OFC.

**Figure 3 children-09-01872-f003:**
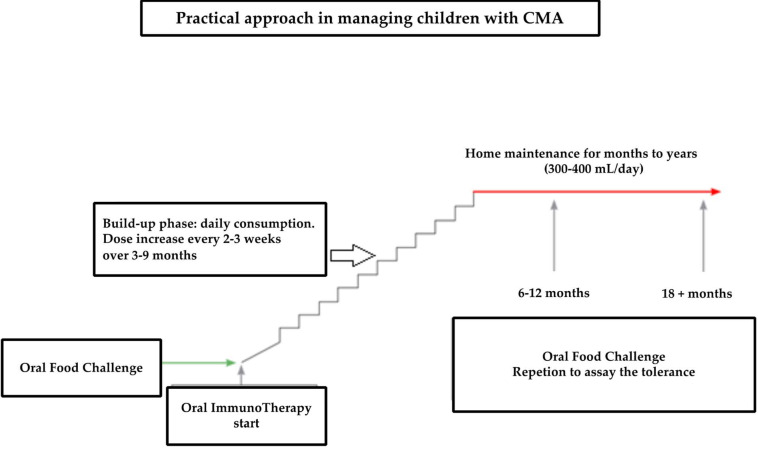
Practical approach in managing CMA.

**Table 1 children-09-01872-t001:** Laboratory data before oral food challenge.

Eosinophilia (>500/μL)	3 (4.0%)
IgE Total (kU/mL)	628.9 ± 1146.20
IgE to casein (kUA/mL)	10.6 ± 20.20
IgE to α-lactoalbumin (kUA/mL)	6.1 ± 15.04
IgE to β-lattoglobulina (kUA/mL)	4.2 ± 9.33
IgE to raw milk (kUA/mL)	14.1 ± 23.62

**Table 2 children-09-01872-t002:** Laboratory data at the last follow-up during OIT.

Eosinophilia (>500/μL)	1 (2.7%)
IgE Total (kU/mL)	820 ± 967.85
IgE to casein (kUA/mL)	11.9 ± 25.41
IgE to α-lactoalbumin (kUA/mL)	6.1 ± 17.26
IgE to β-lattoglobulina (kUA/mL)	3.3 ± 9.9
IgE to raw milk (kUA/mL)	15.8 ± 25.36

**Table 3 children-09-01872-t003:** Comparison of the laboratory between patients without reactions and with reactions during OIT.

	No Reaction	Reaction	*p*
IgE Total (kU/mL)	646.6 ± 1212.07	499.6 ± 655.88	0.97
IgE to casein (kUA/mL)	5.2 ± 11.82	38.6 ± 34.11	<0.001
IgE to α-lactoalbumin (kUA/mL)	3.3 ± 5.69	26.9 ± 35.78	<0.01
IgE to β-lattoglobulina (kUA/mL)	3.3 ± 5.69	13.6 ± 21.86	0.051
IgE to raw milk (kUA/mL)	9.9 ± 19.68	42.3 ± 34.63	<0.001

## Data Availability

The data are available on request.

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
