# Peer review of "Oral Immunotherapy for Children with Cow’s Milk Allergy: A Practical Approach"

_children, 2022, doi:10.3390/children9121872_

Round 1
Reviewer 1 Report
The Conclusions section should be more elaborate, emphasizing your contribution to science.
I suggest adding some pictures with skin manifestations of CM allergy that you noticed during the study.
Author Response
We would thank the Reviewers for helpful comments and suggestions.
The Conclusions section should be more elaborate, emphasizing your contribution to science.
R Many thanks for this comment. We revised the conclusion section as suggested.
I suggest adding some pictures with skin manifestations of CM allergy that you noticed during the study.
R Many thanks for this comment. We provided the picture of a subject with mucocutaneous symptoms.
Reviewer 2 Report
The manuscript published by Tosca A. et all (children-2004824) requires a thorough improvement of the merits An important linguistic correction is also needed.
· Several studies have shown that the IgE levels, measured by skin test and/or serum assay, can discriminate between allergy persistence and future tolerance [12].
In Shek manuscript serum IgE content was measured not a skin prick test. How to perform IgE testin using skin tests????
· pathogenic mechanisms- should be ‘pathological mechanisms’ I believe
· diriment ???
· Bakery products with milk as ingredients are less allergenic, so that they can be used in cases of severe milk reactions. Biblio ?
Yes references needed but it is not right way of thinking anymore. It is confirmed that that is not proven for caseines.
· Defiites –‘deficiences’ I believe
Materials and Methods
· The total serum IgE normal level is > 100 kU/mL. Sensitization is defined when the value of allergen-specific IgE is > 0.35 kUA/mL.
Noo It is not > but < and if we will take into account the limits for age limits are much lower than <100. For children it is respectively
· newborns < 1,5 IU/ml.
· – 12 mounth < 15 IU/ml.
· 1 – 5 year < 60 IU/ml.
· 5 – 9 year < 90 IU/ml.
· The inclusion and exclusion of cases is not described sufficiently and contains a lot of mistakes. It is also carelessly prepared manuscript.
· Residual results, without proper statistical analysis without advanced data analysis.
· Obtained exact numbers of approvals of bioethical committees and informed consents should be carefully described here and informed consents submitted.
· Laboratory data included peripheral eosinophils, total serum IgE, and allergen-specific IgE to casein, a-lactalbumin, b-lactoglobulin, and raw milk.
It is not clear if that data were obtained from database or contents were tested again
Results Discussion Conclusion
· If 40 % of 76 patients needed to stop desensitization and Seventeen patients (30%) were still continuing OIT do we really can conclude that SOI with applied products and protocols should be recommended so far and that the outcomes were favorable?
Maybe it is worth to introduce other formulas (plenty of protocols and formulas for OIT has been described so far) and it could be good to introduce it. It should be definitely discussed though because the discussion contains 5 manuscripts and even the reference here is old.
In my opinion this manuscript is not ready to be published.
Author Response
We would thank the Reviewers for helpful comments and suggestions.
The manuscript published by Tosca A. et all (children-2004824) requires a thorough improvement of the merits An important linguistic correction is also needed.
- Several studies have shown that the IgE levels, measured by skin test and/or serum assay, can discriminate between allergy persistence and future tolerance [12]. In Shek manuscript serum IgE content was measured not a skin prick test. How to perform IgE testin using skin tests????
R Many thanks for this comment. Really, we forgot to report even ref. 9 and 10. Moreover, we provided additional references concerning the role of allergen-specific IgE.
- pathogenic mechanisms- should be ‘pathological mechanisms’ I believe
R Many thanks for this comment. Really, we intended to use the term pathogenetic. Amended
- diriment ???
R Many thanks for this comment. We changed the term.
- Bakery products with milk as ingredients are less allergenic, so that they can be used in cases of severe milk reactions. Biblio ? Yes references needed but it is not right way of thinking anymore. It is confirmed that that is not proven for caseines.
R Many thanks for this comment. We provided the reference and changed the sentence accordingly.
- Defiites –‘deficiences’ I believe
R Sorry, but the term “defiites” does exist in the manuscript. We used the term deficits. Anyway, we changed with deficiency.
Materials and Methods
- The total serum IgE normal level is > 100 kU/mL. Sensitization is defined when the value of allergey specific IgE is > 0.35 kUA/mL.
Noo It is not > but < and if we will take into account the limits for age limits are much lower than <100. For children it is respectively
- newborns < 1,5 IU/ml.
- – 12 mounth < 15 IU/ml.
- 1 – 5 year < 60 IU/ml.
- 5 – 9 year < 90 IU/ml.
R Many thanks for the comment. We amended the sentence accordingly.
- The inclusion and exclusion of cases is not described sufficiently and contains a lot of mistakes. It is also carelessly prepared manuscript.
R Many thanks for this comment. We rephrased the sentences providing more details.
- Residual results, without proper statistical analysis without advanced data analysis.
R Many thanks for this comment. The present manuscript essentially reports the practical experience of a third-level pediatric allergy clinic. As a result, the statistics is eminently descriptive. Further studies will examine in depth some aspects (e.g., the predictive value of cut-off for OIT response…).
- Obtained exact numbers of approvals of bioethical committees and informed consents should be carefully described here and informed consents submitted.
R Many thanks for this comment. We provided the requested information and attached the form used in clinical practice.
- Laboratory data included peripheral eosinophils, total serum IgE, and allergen-specific IgE to casein, a-lactalbumin, b-lactoglobulin, and raw milk. It is not clear if that data were obtained from database or contents were tested again.
R Many thanks for this comment. The lab data derived from the electronic database; reported in the text.
Results Discussion Conclusion
- If 40 % of 76 patients needed to stop desensitization and Seventeen patients (30%) were still continuing OIT do we really can conclude that SOI with applied products and protocols should be recommended so far and that the outcomes were favorable?
R Many thanks for this comment. These data concern the OIT procedure followed by 57 patients and not 76. Moreover, 17 children temporarily interrupted the OIT treatment, only six discontinued OIT. We stated (in the Results) that at the last OIT follow-up…”36 patients (63.2%) were taking milk regularly in the diet without presenting symptoms, 9 (15.8%) preferred to avoid milk, but were taking foods labeled 'may contain traces of milk” without problems, 8 (14%) preferred to avoid milk and foods labeled 'may contain traces of milk”, 4 (7%) were lost to follow-up.”…,. We revised the conclusions accordingly.
Maybe it is worth to introduce other formulas (plenty of protocols and formulas for OIT has been described so far) and it could be good to introduce it. It should be definitely discussed though because the discussion contains 5 manuscripts and even the reference here is old.
R Many thanks for this comment. We revised the Discussion. However, the Discussion included 6 references, five published in 2022 and one in 2019. However, as already mentioned in previous answers, this manuscript merely reports the experience of a centre in the context of a topic that has not yet been definitively standardised.
In my opinion this manuscript is not ready to be published.
Informed Consent Form
U.O.S.D. di Centro Malattie Allergiche
Direttore M.A.Tosca
Test di Provocazione Orale con Alimenti TPO
Che cos’è il “Test di provocazione orale con alimenti”
- Per TPO si intende la somministrazione orale di un alimento in soggetto con sospetta allergia alimentare, eseguita sotto controllo medico, in modo standardizzato e controllato.
- E’ l’unico esame che consente di porre con certezza diagnosi di allergia alimentare, discriminando la patologia dalla semplice sensibilizzazione e quindi fornisce la possibilità di prescrivere correttamente una dieta di eliminazione rigorosa verso l’alimento causale.
- Inoltre è un test essenziale nel follow-up di bambini con allergia alimentare, soprattutto per latte vaccino e uovo, per verificare nel tempo l’acquisizione di uno stato di tolleranza verso l’alimento.
Metodica
Come il paziente viene preparato all’esame
Per essere sottoposto al test di provocazione orale, il paziente deve essere a digiuno almeno da 6 ore prima della procedura per gli alimenti e 4 ore per i liquidi chiari (acqua, the, camomilla etc). Inoltre, è opportuno sospendere l’assunzione di: Broncodilatatori a breve durata d'azione da almeno 8-12 ore; Broncodilatatori a lunga durata d'azione da almeno 24 ore, Antileucotrienici da almeno 24 ore. Verrà posizionato un accesso venoso, per l’eventuale somministrazioni di farmaci.
Come si esegue il test
Un test di provocazione orale (TPO) può essere eseguito in diversi modi:
Eseguiamo 2 tipologie di TPO:
- TPO in aperto: Si somministra l’alimento nella sua forma naturale e sia il bambino (e i suoi genitori) che il Medico conoscono la natura dell’alimento somministrato. È il test più semplice, con minori costi, ma non esclude la componente psicologica (falsi positivi con sintomi soggettivi o rifiuto). Tale esame comunque, soprattutto in età pediatrica e nel sospetto di allergia alimentare IgE-mediata con sintomi obbiettivi, costituisce il test di prima istanza.
- TPO in singolo cieco: Si somministra l’alimento in forma “nascosta” ma solo il bambino (e i suoi genitori) non sanno se sta assumendo l’alimento sospettato di causare allergia o il placebo, mentre il medico ne è a conoscenza. Si esclude così la componente psicologica dei genitori e/o del paziente, ma non dell’operatore, da cui ne consegue che vi possano essere dei dubbi di interpretazione da parte del medico per sintomi non obbiettivi
Procedura
Prevede: a) l’esame obbiettivo accurato (frequenza cardiaca e respiratoria e pressione arteriosa); b) il posizionamento di un accesso venoso (ago-cannula) ; c) la preparazione dei farmaci di primo impiego per eventuali reazioni avverse (antistaminico, adrenalina, steroide); d) inizio somministrazione e registrazione dati (ora/quantità di alimento ad ogni dose/sintomi soggettivi e oggettivi). In genere le dosi iniziali consigliate, diverse per i diversi alimenti, vengono aumentate gradatamente del doppio o del triplo, ogni 15-20 minuti, fino ad arrivare ad una dose massimale corrispondente circa alla dose assunta nella quotidianità. Ad ogni successiva introduzione vanno rivalutate le condizioni del paziente e registrata la comparsa di eventuali sintomi obbiettivi o soggettivi. Il TPO termina quando viene somministrata la dose massimale in assenza di sintomi o in caso di comparsa di sintomi obbiettivi. In caso di sintomi solo soggettivi, come nausea, dolore addominale, prurito cutaneo o fastidio al cavo orale si può, in caso di stabilizzazione, proseguire il TPO ripetendo la stessa dose di alimento e allungando l’intervallo delle dosi.
Al termine del TPO si raccomanda:
- In assenza di reazioni, un tempo di osservazione di almeno 2-4 ore dall’ultima dose
- In caso di reazioni un tempo di osservazione ospedaliera prolungata da 6 ore, fino ad almeno 24 ore in caso di anafilassi.
Dove sarà effettuato l’esame
Il test verrà eseguito presso il Reparto di Degenza o nei locali del Day Hospital della U.O.C. di Pediatria a Indirizzo Pneumologico e Allergologico, rispettivamente al Padiglione 3 terra o al 2° piano del Padiglione 20, “Ospedale di Giorno”. Generalmente l’esame viene eseguito al mattino. Nei locali sono presenti e disponibili durante tutta la durata della prova di scatenamento alimentare orale e durante il periodo di osservazione tuttele attrezzature necessarie per il trattamento delle reazioni anafilattiche come lo shock.
Chi pratica l’esame
Un Dirigente Medico della U.O.C. di Pediatria a Indirizzo Pneumologico e Allergologico, coadiuvato da una Infermiera Professionale di comprovata esperienza, sempre pronti nel riconoscere e trattare immediatamente gli eventuali sintomi clinici avversi.
Chi pratica l’esame
Un Dirigente Medico della U.O.C. di Pediatria a Indirizzo Pneumologico e Allergologico, coadiuvato da una Infermiera Professionale di comprovata esperienza, sempre pronti nel riconoscere e trattare immediatamente gli eventuali sintomi clinici avversi.
Indicazioni
Le indicazioni sono le seguenti:
- Verificare la diagnosi di allergia alimentare in tutti i casi in cui il quadro clinico è dubbio sia in presenza che in assenza di IgE specifiche, eventualmente anche per liberalizzare una dieta di eliminazione ritenuta incongrua.
- Verificare lo stato di tolleranza verso un alimento mai introdotto in presenza di IgE specifiche per quell’alimento.
- Verificare nel tempo l’eventuale acquisizione di tolleranza verso l’alimento
Controindicazioni
Criteri di esclusione all’esecuzione del TPO sono:
- Anamnesi di recente anafilassi verso l’alimento causale (entro 12 mesi)
- Mancata sospensione di farmaci che possono interferire con la risposta del test, come gli antistaminici e gli steroidei sistemici (almeno 10 giorni)
- Condizioni cliniche non stazionarie od ottimali
- Pazienti con assenza di sintomi all’assunzione accidentale dell’alimento in causa (tolleranza già dimostrata)
Rischi e complicazioni.
La reintroduzione di alimenti può comportare possibili rischi e reazioni locali o sistemiche, durante e dopo il test, tra le quali le più frequenti sono:
- Manifestazioni cutanee (orticaria, angioedema, rush con prurito, orticaria)
- Sintomi al cavo orale assimilabili alla sindrome orale-allergica (prurito al cavo orale, rigonfiamento delle labbra, della lingua, del palato, difficoltà alla deglutizione)
- Manifestazioni gastroenteriche (nausea, vomito, diarrea)
- Manifestazioni respiratorie (rino-congiuntivite, tosse, dispnea e broncospasmo, edema della glottide)
- Manifestazioni sistemiche (anafilassi fino allo shock anafilattico)
Ricordare che …
- I pazienti e/o i loro genitori non devono mai eseguire autonomamente un test al proprio domicilio.
- Se il paziente ha una storia di reazioni allergiche gravi dopo l'ingestione di alimenti, va fornita una specifica consulenza sotto forma di un piano di emergenza scritto con i trattamenti del caso.
- Inoltre, il paziente va educato su come assumere farmaci di emergenza (ad esempio, adrenalina iniettabile, antistaminici, ecc) atto a gestire al meglio l'eventualità di una crisi anafilattica pericolosa per la vita del soggetto.
- Inoltre, se il caso esiste, vanno incoraggiati i pazienti a custodire e portare con i farmaci di emergenza sempre, in modo che in qualsiasi momento essi dovessero servire devono essere prontamente disponibili.
Misure di sicurezza:
La presenza di un rischio è ovviamente intrinseca nell’esecuzione di ogni test di tolleranza, pertanto nell’effettuare un TPO è necessario mettere in atto tutti gli accorgimenti possibili allo scopo di essere nelle condizioni ottimali per poter trattare tempestivamente ed adeguatamente eventuali reazioni anche gravi. In particolare con pazienti a rischio elevato (precedenti reazioni avverse gravi, sensibilizzazione elevata, co-morbidità con asma instabile, età del paziente, tipo di alimento testato es. frutta secca) il test deve essere eseguito sempre con la collaborazione di un rianimatore pre-allertato.
Consenso informato
1. Dichiaro di aver compreso quanto sopra esposto
Pertanto:
- Acconsento all’attuazione del programma diagnostico (Provocazione Orale con Alimenti) secondo quanto sopra proposto. a mio/a figlio/a …………… nato/a a …………., il ………..
affetto da ……………………………………………………………………………………………..
Il padre:
Cognome ____________________________ Nome ____________________ Firma ____________
La madre:
Cognome ____________________________ Nome ____________________ Firma ____________
Motivazioni per cui non possibile la presenza di entrambi i genitori : ……………………………….
……………………………………………………
Il Genitore presente dichiara che il coniuge assente è informato ed acconsente all’esecuzione della metodica
Firma ________________________
Il tutore:
Cognome ____________________________ Nome ____________________ Firma ____________
Firma del Medico ________________________
Firma dell’Operatore Sanitario ________________________
data ___________
Round 2
Reviewer 2 Report
Despite some aspects do not convince me but some of previous failures has been corrected. I believe the manuscript can be accepted in present form.
Author Response
Despite some aspects do not convince me but some of previous failures has been corrected. I believe the manuscript can be accepted in present form.
R Many thanks for your consideration.